# Long-Term Memory Function Impairments following Sucrose Exposure in Juvenile versus Adult Rats

**DOI:** 10.3390/biomedicines10112723

**Published:** 2022-10-27

**Authors:** Héctor Coirini, Mariana Rey, María Claudia Gonzalez Deniselle, María Sol Kruse

**Affiliations:** 1Laboratorio de Neurobiología, Instituto de Biología y Medicina Experimental-CONICET, Vuelta de Obligado 2490, Buenos Aires C1428ADN, Argentina; 2Laboratorio de Bioquímica Neuroendócrina, Instituto de Biología y Medicina Experimental-CONICET, Vuelta de Obligado 2490, Buenos Aires C1428ADN, Argentina; 3Departamento de Fisiología, Facultad de Medicina, Universidad de Buenos Aires, Paraguay 2155, Buenos Aires C1121ABG, Argentina

**Keywords:** NOR, medial prefrontal cortex, hippocampus, RAGE, glucose tolerance

## Abstract

We previously described that excessive consumption of sucrose during youth produces fear memory and anxiety-like behavior in adulthood. Here, we evaluated whether high cognitive function is also affected by studying early sucrose consumption in object recognition memory (NOR). Male Sprague Dawley rats were tested for short-term, long-term, and consolidated NOR after 25 days of unlimited sucrose access in juvenile (PD 25–50) or adult age (PD 75–100). All rats spent equal time exploring the two objects during the sample phase T1. When animals were exposed for 2, 24 h or 7 days later to a copy of the objects presented in T1 and a novel object, the sucrose-exposed juvenile group failed to distinguish between the familiar and the novel objects in contrast with the rest of the groups. Sucrose-exposed animals developed hypertriglyceridemia and glucose intolerance, but juvenile animals showed increased fasting glycemia and sustained the glucose intolerance longer. Moreover, sucrose decreased hippocampal proBDNF expression in juveniles while it was increased in adults, and sucrose also increased RAGE expression in adults. The NOR exploration ratio correlated negatively with basal glycemia and positively with proBDNF. Taken together, these data suggest that sucrose-induced alterations in glucose metabolism may contribute to a long-term decline in proBDNF and impaired recognition memory.

## 1. Introduction

Sugar intake in modern society has increased dramatically over the past years [1] together with obesity and metabolic disturbances [2]. This augmentation is largely attributable to the rising consumption of sugar-sweetened beverages (SSBs), the largest single source of added sugar consumption worldwide [3]. Overconsumption of SSBs is directly associated with the development of obesity, type 2 diabetes, and cardiovascular disease in large epidemiological studies [4,5,6]. Moreover, emerging research indicates that SSBs impair brain functioning even in the absence of extreme weight gain or excessive energy intake [7,8,9].

Disturbances within the hippocampus (HIP) occur after consumption of SSBs. Animals exposed to SSBs present reduced brain-derived neurotrophic factor (BDNF) mRNA expression and increased inflammation in the HIP [8,10,11,12]. BDNF is a signaling molecule that is related to synaptic plasticity [13] and energy metabolism [14] to promote the survival, maintenance, and growth of neurons [15]. BDNF and its precursor, proBDNF, are key modulators of memory consolidation/reconsolidation and learning [16,17].

High-sucrose consumption has been shown to induce glucose intolerance and hyperglycemia in different animal models [10,18,19,20,21]. Hyperglycemia increases the advanced glycation end products (AGE) and its receptor (RAGE) activation, induces oxidative stress, and increases the expression of inflammatory mediators [22,23,24,25]. The activation of the AGE-RAGE pathway has been strongly associated with diabetes-related complications [26] and cognitive impairments in Alzheimer’s disease [27]. More recently, increases in AGE and oxidative stress have been found in prediabetic patients [28], and it has been associated with abnormal behavior in the offspring born to diabetic dams [29] and with memory impairments in female rats subjected to a high-fructose diet [30]. In vitro studies also show increased RAGE expression and RAGE-dependent alterations in the HIP of hyperglycemic mice [31].

Taken together, these studies suggest that the mechanisms of high-sucrose diets for inducing cognitive deficits may include hyperglycemia and increased expression/activation of RAGE.

Many studies have shown that daily access to sucrose, either intermittently or continuously, disrupts place recognition memory in both young and adult animals, indicating that SSBs impair hippocampal-dependent functions [8,32,33]. However, there are conflicting data on whether SSBs consumption affects object recognition memory.

The novel object recognition (NOR) test measures non-spatial memory and the test relies on the rats’ innate preference to explore novelty [34]. Rats with lesions to the perirhinal cortex show deficits in object recognition in short-term memory [35], whereas damage to the HIP can also impair object recognition memory at longer retention intervals of 24 h [36]. Chronic sucrose ingestion (32% sucrose solution for 8 weeks) alters NOR in Long–Evans rats when retention intervals were extended to 1 h, indicating that perirhinal cortex function is also comprised by sugar consumption [37]. In female rats, fructose consumption also alters NOR test performance with retention intervals of 2 h and 24 h [30]. Furthermore, intermittent sucrose consumption in juvenile animals affects NOR only when the objects shared multiple characteristics [33,38], whereas in adults, continuous sucrose consumption does not affect NOR [8,39]. Together, these data indicate that the amount of sucrose consumed and the age of exposure to this consumption are probably related to the cognitive deficits found.

Limited information is available about the potentially detrimental long-term effects on cognitive development of juvenile individuals exposed to high sucrose consumption [9,40,41]. This topic is of great interest considering that SSBs consumption has increased dramatically among children and adolescents over recent years [42,43,44] and this period of life is characterized by rapid brain development and learning of new information and skills [45]. Therefore, using a rat a model of childhood–adolescent (youth period) SSB consumption, we investigated whether impairments in perirhinal cortex/ hippocampal-dependent memory function due to early-life SSB access persist well into adulthood. The long-term effects of unlimited access to 10% sucrose solution during youth on the NOR task were examined using the two-bottle-choice paradigm. Following this protocol we have recently demonstrated that sucrose-exposed animals consume only 5–10% of water [7], a similar value found in the infant–juvenile population of Argentina [44,46], showing that the consumption of SSBs (juices, sodas, etc.) has become naturalized over water on a daily bases [47,48,49].

To examine some of the potential neurobiological mechanisms for cognitive deficits, we studied the impact of sucrose overconsumption on some metabolism parameters, including basal glycemia (BG), glucose tolerance test (GTT), triglycerides (TG), and corticosterone. We also investigated the long-term effect of sucrose consumption on the HIP and medial prefrontal cortex (mPFC) expression of the oxidative stress marker RAGE and the HIP proBDNF by Western blot. We chose these brain areas because we found that they are especially sensitive to sucrose consumption during youth and because they are key structures that contribute to object recognition memory [7,50,51,52,53].

In parallel, the effects of sugar intake on adult rats were also analyzed in order to assess whether the behavioral changes produced by an excessive consumption of sucrose are due to a specific action during youth, or if it also produces similar effects in adulthood.

## 2. Materials and Methods

### 2.1. Experimental Animals

Animal procedures were approved by the Animal Care and Ethical Use Committee of the Institute of Biology and Experimental Medicine, IBYME, Argentina (protocol numbers 010/2016 and 33/2019), in accordance with guidelines defined by European Community Council Directive (86/609/EEC) and the National Institutes of Health for the Care and Use of Laboratory Animals. Sprague–Dawley male rats were maintained on a 12 h light:12 h darkness cycle with food and water available ad libitum. Over a 25 day period, 25 day old rats (juvenile group) and 75 day old rats (adult group) had unrestricted access to tap water (control groups) or the choice to drink a 10% sucrose solution (*w*/*v* made up of tap water, sucrose groups) ad libitum. Fresh sucrose solution was prepared every second day. After this 25 day period of sucrose exposure, the bottles containing a 10% sucrose solution were removed and all animals drank only water for other 25 days (Figure 1) [7]. A total of 10 animals were randomly assigned per group and at the end of the protocol, all animals were tested for NOR (days 1–7, Figure 1). After the test, on day 8, animals were euthanized by decapitation.

### 2.2. Triglycerides and Corticosterone Serum Levels

Animals were fasted for 6 h and blood samples were taken from the tail vein (∼0.5 mL) or from the trunk after decapitation, and immediately centrifuged for 5 min at room temperature (RT) at 1800× *g* in a tabletop centrifuge. The serum was collected and the levels of TG were measured by spectrophotometry (Wiener Labs S.A.I.C., Rosario, Argentina) as described previously and corticosterone levels were determined by radioimmunoassay using a specific antibody [54,55,56].

### 2.3. Glucose Tolerance Test

Immediately after finishing the sucrose period and 25 days after that (Figure 1) the animals were fasted for 6 h. Blood samples were taken from the tail vein and fasting glucose levels were determined using a commercial strip and a glucometer (OneTouch Ultra, Johnson & Johnson, CABA, Argentina) as previously performed [54]. A glucose overload was administered by i.p. injection (2 g/kg body weight) and blood glucose levels were measured at 30 min, 60 min, and 120 min post-injection. The area under the glucose curve (AUC) during the GTT was calculated using the Graph Pad-Prism Software (Graph Pad Software Inc., v. 6.01, San Diego, CA, USA), and included the baseline glucose measurement.

### 2.4. Novel Object Recognition Task

The test apparatus consisted of a dark open box made of wood (75 cm length × 30 cm height × 55 cm width) that was illuminated by a 42 W light suspended 100 cm above the box facing the ceiling. The light intensity was equal in the different parts of the apparatus. The objects to be discriminated differed in shape and texture, made of glass, plastic, ceramic, or metal in four different shapes: glass jars, ceramic mugs, metallic cylinders, and plastic cubes and could not be displaced by rats [57,58]. Briefly, during the week before the test, the rats were handled once a day for 3 consecutive days. At 5 days before testing, the rats were allowed to explore the apparatus for 6 min each day. During these habituation sessions no objects were present in the box. A total of 24 h after the last habituation session, the rats were trained for object recognition by allowing them to explore for 6 min two identical samples (objects) placed in the test arena (sample phase). During this sample phase (T1), the objects (two orthogonal glass jars 12.5 cm high and 5 cm in diameter) were placed in two opposite corners of the apparatus in a random fashion, 10 cm from the side walls. A rat was placed in the middle of the apparatus and allowed to explore the two identical objects. After T1, the rat was returned to its home cage and after an intertrial interval (ITI) of 2 h (short-term memory, T2), 24 h (long-term memory, T3), and 7 days after the sample phase (consolidated memory, T4), the rats were tested again for the “choice” trials performance. During T2, T3, and T4 a novel object replaced one of the objects presented during T1. Accordingly, the rats were re-exposed to two objects for 6 min: a copy of the familiar (F) object and the novel (N) object (i.e., one glass jar was exchanged for a rough yellow metal cylinder 9.5 cm high and 5.5 cm in diameter, an orange plastic box 10.5 cm long × 9 cm wide × 4 cm high, and a conical cup of white ceramic 8.5 cm high and 8.5 cm in diameter at T2, T3, and T4, respectively). All combinations and locations of the objects were counterbalanced to reduce potential bias caused by preference for particular locations or objects. Sessions T1–T4 were video-recorded and analyzed for line crossing with the ANY-maze© software (Stoelting Co., Wood Dale, IL, USA). The time spent by the rats exploring each object during T1–T4 was manually recorded with a stopwatch by the experimenter and a second investigator blind to the group assignment. Exploration was defined as follows: directing the nose toward the object at a distance of 2 cm or less and/or touching the object with the nose. Turning around or sitting on the object was not considered exploratory behavior. Rats should explore at least 20 s (for both objects) in each trial to be included for the analysis. The correlation between the two observers was high (r > 0.90). Data were reported as an exploration ratio, which was the time spent exploring the novel object divided by the time spent exploring both objects [tN/(tN + tF)].

### 2.5. Western Blot

Sprague–Dawley rats were killed by decapitation and samples of the mPFC and the HIP were rapidly dissected out using the Paxinos rat brain atlas as a reference guide [59], quickly frozen on dry ice, and stored at −80 °C as previously performed [60,61]. Homogenates were prepared by sonication in ice-cold lysis buffer (50 mM Tris–HCl, 150 mM NaCl, 2 mM EDTA, 1 mM phenylmethylsulphonylfluoride, 1 mM Na3VO4, and 1% Triton 100, pH 7.4) containing a protease inhibitor cocktail (Roche Diagnostics, CABA, Argentina) [62,63]. A total of 20 µg of protein was separated by 10% SDS–PAGE in Tris–glycine electrophoresis buffer at 120 V for 90 min. Proteins from gels were transferred onto PVDF membranes (Bio-Rad, CABA, Argentina), and the membranes were blocked with TBS-T (20 mM Tris, pH 7.5; 150 mM NaCl; and 0.1% Tween-20) containing 5% fat-free milk for 1 h. Blocked membranes were incubated with the primary antibody in TBS-T containing 5% fat-free milk at 4 °C overnight. The primary antibodies used were RAGE (mouse, ref #sc-365154), proBDNF (mouse, ref #sc-65514), and F-actin (goat, ref #sc-1616) all from Santa Cruz Biotechnology (CABA, Argentina), 1:500 dilution. Immunoblots were then washed with TBS-T three times and incubated at RT for 1 h with the respective HRP-conjugated secondary antibodies (goat anti-mouse, 1:1000 dilution, ref #170-6516, BioRad, and donkey anti-goat, 1:2000 dilution, ref #sc-2020, Santa Cruz Biotechnology). Chemiluminescence was detected with the ECL system (GE Healthcare Life Sciences, CABA, Argentina) and exposure to hyperfilm (GE Healthcare Life Sciences). All membranes were then stripped and reprobed for F-actin as a loading control. Signals in the immunoblots were scanned and analyzed by Scion Image Software (National Institutes of Health, Washington, DC, USA). The amount of target protein was indexed to F-actin in all cases to ensure correction for the amount of total protein on the membrane. The results are reported as percentages of values obtained from the expression of target proteins compared with controls.

### 2.6. Statistical Analysis

The distribution normality of variables was assessed using D’Agostino–Pearson, Shapiro–Wilk, and Kolmogorov–Smirnoff normality tests (Graph Pad-Prism, Graph Pad Software Inc., v. 6.01, San Diego, CA, USA). The significances between variables were evaluated by three-way ANOVA (treatment, age, left/right object), two-way ANOVA (treatment, age), or two-way repeated measures ANOVA (RM ANOVA; time, treatment, age; all factors are indicated for each assay in the results section), using the IBM^®^ SPSS^®^ Statistics 21 Software or the Statview Statistical Software (SAS Institute, Inc., Cary, NC, USA; v5.0.1) followed by Tukey’s post hoc test. Each test was selected to estimate changes in a quantitative variable according to three or two independent variables or to understand if there is an interaction between two factors determined at different times on the dependent variable. The correlation analyses were calculated by simple linear regression using Statview Statistical Software. Significance was assumed at *p* < 0.05. Values are reported as the mean and the standard deviation (SD) from two independent experiments.

## 3. Results

### 3.1. Long-term Effects of Sucrose Exposure on Learning

Preferential exploration of the novel versus the familiar object provided a measure of recognition memory comprising three basic processes: encoding, consolidation, and retrieval. After the exposure to two identical objects (sample phase, T1), memory was evaluated in the following trials by exchanging one of the familiar objects for a new one.

The rats of the control and sucrose-exposed groups spent the same time exploring the two objects during the sample phase T1 (three-way ANOVA, FTreatment-Age-Left/Right Object (16,480) = 0.004; *p* = 0.984; Figure 2A). However, differences by age were detected showing that juvenile animals spent more time exploring the identical objects (three-way ANOVA, FAge (1,64) = 21.573; *p* < 0.0001; Figure 2A). After 2 h of ITI, all groups were exposed to one of the objects presented in T1 and a novel object (short-term memory, T2) and the exploration ratio (tN/(tN + tF) was calculated. There was a significant main effect of age-treatment interaction on the ability to discriminate the novel from the familiar object (two-way ANOVA, FInteraction (1,33) = 4.208; *p* = 0.0497). Both the adult and juvenile control groups and the sucrose-exposed adult group exhibited a clear preference for exploring the novel object, in contrast with the sucrose-exposed juvenile group (Tukey’s post hoc test, control vs. sucrose of the juvenile group *p* = 0.024, and juvenile-sucrose vs. adult-sucrose *p* < 0.0001; Figure 2B).

Similarly, two-way ANOVA showed differences in the preference to explore the novel object in relation to the familiar one at 24 h (long-term memory, T3; FInteraction (1,33) = 11.908; *p* = 0.0015) and 7 days after the sample phase (consolidated memory, T4; FInteraction (1.33) = 4.932; *p* = 0.0370). Both the adult and juvenile control groups and the sucrose-exposed adult group exhibited higher exploration ratios in T3 and T4 compared with the sucrose-exposed juvenile group, indicating that the ability to distinguish between the novel and the familiar objects at 24 h and 7 days ITI was also impaired in this last group (Tukey’s post hoc test, 24 h: control vs. sucrose of the juvenile group *p* = 0.033 and juvenile-sucrose vs. adult-sucrose *p* < 0.005; Figure 2C and 7 days: control vs. sucrose of the juvenile group *p* = 0.034 and juvenile-sucrose vs. adult-sucrose *p* < 0.001; Figure 2D).

Importantly, sucrose-exposed animals exhibited identical locomotor activity compared with their respective age controls as indicated by the similar number of line crossings during the first trial, T1, where the animal is free to explore two identical objects (two-way ANOVA, FTreatment-Age(1,33) = 0.002; *p* = 0.963; Figure 3A). In fact, the comparison of locomotor activity during the T1–T4 NOR trials revealed no significant interaction between trial, age, and treatment (two-way RM ANOVA, FInteraction (3,99) = 0.352; *p* = 0.788; Figure 3B). Similarly, total exploration time did not differ between control and sucrose groups neither on T1 (two-way ANOVA, FTreatment-Age (1,33) = 0.030; *p* = 0.863; Figure 3C) nor during T1–T4 NOR trials (two-way RM ANOVA, FTreatment-Age-Trial (3,99) = 0.578; *p* = 0.631; Figure 3D). However, significant differences were found by age in the number of line crossings (two-way RM ANOVA, FTrial-Age (3,99) = 6.298; *p* = 0.001) and in the total exploration time (two-way RM ANOVA, FTrial-Age (3,99) = 5.785; *p* = 0.001). A subsequent post hoc analysis revealed that the juvenile group crossed a greater number of lines at T1 (Tukey’s test, *p* < 0.005; Figure 3A,B) and explored more at T1 and T3 (Tukey’s test, *p* < 0.0001 and *p* < 0.010, respectively; Figure 3C,D). Altogether, these results indicate: (1) Both adult and juvenile control groups and the sucrose-exposed adult group kept in memory the familiar object and could distinguish it from the novel one at least 7 days after a single sample phase; (2) sucrose treatment does not interfere with the locomotor activity of the animals habituated to the arena; and (3) the impaired learning performance of the sucrose-exposed juvenile group is not due to an attention deficit but likely results from an impaired memory consolidation and/or reconsolidation.

### 3.2. Short- and Long-term Effects of Sucrose Exposure on Glucose Tolerance Test and Serum Levels of BG, TG, and Corticosterone

The ability to regulate glucose overload was tested after 25 days of sucrose consumption in juvenile and adult group animals (PD50 and PD100, respectively; Figure 1). Rats from the sucrose group, both juveniles and adults, showed higher glycemia levels than their controls of age, but these differences were not significant when the GTT curves were analyzed together (two-way RM ANOVA, FTime-Treatment-Age (3,99) = 1.776; *p* = 0.156; Figure 4A). Only differences by age were found (two-way RM ANOVA, FTime-Age (3,99) = 17.041; *p* < 0.0001; Figure 4A). However, the AUC was significantly higher by sucrose treatment in both age groups (two-way ANOVA, FTreatment (1,33) = 7.658; *p* = 0.0091; Figure 4B). Moreover, the comparison of AUC values revealed differences by age (two-way ANOVA, FAge (1,33) = 16,954; *p* = 0.0002; Figure 4B).

Twenty-five days after the end of the sucrose consumption period the GTT was repeated (PD75 and PD125, juvenile and adult group, respectively; Figure 1). The two-way RM ANOVA revealed differences by treatment and interaction (FTreatment-Time (3,99) = 4.024; *p* = 0.009 and FTreatment-Age Time (3, 99) = 3.153; *p* = 0.028; Figure 4C). A subsequent analysis demonstrated that only the sucrose-exposed juvenile group showed long-term glucose intolerance (Tukey’s post hoc test, 30 min, 60 min, and 120 min, *p* < 0.019; Figure 4C). Concomitantly, the two-way ANOVA for AUC values revealed differences by treatment (FTreatment (1, 33) = 4.445; *p* = 0.0429; Figure 4D).

Differences in the basal glycemia (BG) were only observed in juvenile groups immediately after finishing the sucrose period (PD50, BG1), but not in the long term (PD75, BG2) (two-way ANOVA, FTreatment (1,33) = 8.681; *p* = 0.006) or in the sucrose-exposed adult group (two-way ANOVA, FTreatment (1,33) = 0.005; *p* = 0.996; Table 1). In addition, BG1 was negatively correlated with novelty preferences in the long term with T2 and T4 (one-way ANOVA, FT2 (1,26) = 5.283; *p* = 0.0302; r^2^ = 0.174 and FT4 (1,26) = 6.144; *p* = 0.0227; r^2^ = 0.244; Figure 5A), whereas BG2 correlated with a lower exploration ratio in all memory trials (one-way ANOVA, FT2 (1,31) = 8.197; *p* = 0.0076; r^2^ = 0.215, FT3 (1,31) = 5.617; *p* = 0.0242; r^2^ = 0.153 and FT4 (1,26) = 5.744; *p* = 0.0264; r^2^= 0.223; Figure 5B).

Serum TG was also measured in all animals immediately after finishing the sucrose treatment and 25 days after this period. Sucrose increased significantly the TG levels in both juvenile and adult animals in the short term (PD50/ PD100) but not in the long term (PD75/PD125) (two-way ANOVA, FTreatment-Time (1,33) = 5.698; *p* = 0.027 and F (1,33) = 12.111; *p* = 0.002, respectively; Table 1). Serum corticosterone levels were measured at the end of the experimental protocol and no differences were observed between the sucrose and control groups in both age groups (two-way ANOVA, FTreatment-Time (1,33) = 0.123; *p* = 0.731) as previously reported [64].

### 3.3. Long-Term Effects of Sucrose on RAGE Expression

Twenty-four hours after T4, animals were euthanized and RAGE expression was evaluated in HIP and mPFC by Western blot. Differences by age were detected in the HIP by two-way ANOVA (FAge (1,16) = 11.361; *p* = 0.004). Rats from the juvenile group presented higher RAGE expression in the HIP compared with rats from the adult group (Tukey’s post hoc test, *p* < 0.005; Figure 6A). Differences by treatment–age interaction were only observed in the mPFC (two-way ANOVA, FTreatment-Age (1,16) = 6.192; *p* = 0.025). The sucrose exposure in the juvenile group did not affect RAGE expression, whereas the same treatment in the adult group raised these values in the long term (Tukey’s post hoc test, *p* = 0.048; Figure 6B). No correlations were found between RAGE expression in the HIP or mPFC and the exploration ratios T2–T4. Only in the HIP, RAGE expression positively correlated with BG1 and BG2 (one-way ANOVA, FBG1 (1,19) = 13.763; *p* = 0.0016; r^2^= 0.433 and FBG2 (1,19) = 4.995; *p* = 0.041; r^2^= 0.250; data not shown).

### 3.4. Long-Term Effects of Sucrose on proBDNF Expression

Two-way ANOVA showed significant differences between the studied groups in the HIP (FTreatment-Age (1,16) = 13.456; *p* = 0.003). Animals from the sucrose-exposed juvenile group showed a decrease in pBDNF levels, whereas animals from the sucrose-exposed adult group showed increased values compared with their respective age controls (Tukey’s post hoc test, *p* = 0.035 and *p* = 0.013, respectively; Figure 7A). When all animals were considered, the proBDNF levels positively correlated with the exploration ratio in the NOR memory tasks T3 and T4 (one-way ANOVA, FT3 (1,19) = 5.470; *p* = 0.0334; r^2^ = 0.268 and FT4 (1,19) = 14.617; *p* = 0.0034; r^2^ = 0.3076 Figure 7B), indicating that higher levels of HIP proBDNF correspond to a better novelty recognition. No correlations were detected between hippocampal proBDNF expression and BG1, but proBDNF expression negatively correlated with BG2 (one-way ANOVA, FBG2 (1,19) = 7.943; *p* = 0.0137; r^2^ = 0.362; Figure 7C). No differences in proBDNF levels were found in the mPFC (data not shown).

## 4. Discussion

In this study, we found that exposure to 10% sucrose during youth produces long-term impairment in object recognition memory, whereas comparable sucrose exposure in adult rats fails to induce such effects. Animals were exposed to three consecutive trials with different delays after the sample phase (2 h, 24 h, and 7-days ITI). For each trial (T2, T3, and T4) a copy of the object presented during the sample phase (T1) and a novel object of different shape, color, and height were introduced in the arena. The novel and the familiar objects were counterbalanced and placed in different positions relative to the location they occupied in the previous trial. When learning performances of control and sucrose-exposed groups were compared, differences in age related to sucrose exposure were noticed. After the sample phase T1, which consisted of time periods of 2 h, 24 h, or 7 days, adult animals from the sucrose-exposed juvenile group failed to distinguish between the familiar and the novel objects. This suggests that early exposure to unlimited sucrose could affect the initial encoding and acquisition of the object, since the same familiar object was presented in the T2–T4 trials. Nonetheless, the difficulty in differentiating the novel object from the familiar one during the T2–T4 trials could also be the expression of impaired memory consolidation or memory retrieval. More studies are required to distinguish between these possibilities, which are not mutually exclusive. Notably, since the short duration between the familiarization of T1 and the test T2 trials is primarily dependent on the perirhinal cortex, these data show that early life sugar consumption impairs perirhinal cortex-dependent memory in addition to hippocampal-dependent memory in adulthood [9,41].

No differences were found between the sucrose-exposed group and its age control in the total time spent exploring the objects neither during the sample phase T1 nor during the T2–T4 trials, indicating that the alterations found in the learning performance of sucrose-exposed juvenile group are not due to an attention deficit but likely result from impaired NOR. Furthermore, the sucrose-exposed animals exhibited identical locomotor activity levels as shown by the similar number of line crossings during the test trials relative to their corresponding age control. Notably, we previously reported anxiety-like behavior measured by a reduced exploration of the center of the open field [7], however, in this study animals were habituated for 5 consecutive days before being tested and the arena dimensions were smaller than the one used in the open field. Altogether, these observations show that animals under the NOR testing conditions do not show anxiety and, consequently, the cognitive defects observed result from learning and memory impairment and not from the interference of stress during the study.

Our results are in accordance with those reporting sugar-induced impairments in NOR memory tasks [30,37,65]. However, most of the studies looked at the immediate effect of sugar on memory, but not whether the effect persists over time. Here, we demonstrate long-term sugar-induced disturbances in animals that had unlimited access to sucrose during youth. In contrast, some studies have shown that perirhinal-dependent memory was not disrupted following either intermittent [32,33] or continuous sucrose consumption in rats [8,39]. However, these latest studies were carried out in adult rats and like our study, no effects were observed in this age group [8,39]. Even more, a recent study in young animals showed that intermittent access to a 10% sucrose solution for 2 weeks can disrupt perirhinal-dependent memory when objects have increased similarity and share multiple features [38]. In this study, female rats consumed significantly more sucrose solution than males and performed worst in the memory tasks, and the authors finally suggest that with a prolonged consumption of sucrose (more than 2 weeks) it could also further decrease the NOR performance in males. Altogether, these data indicate that juvenile animals are more susceptible to the adverse effects of sucrose consumption on cognition. It should be noted that these cognitive changes are observed in the absence of weight differences [7,64,66] suggesting that the metabolic disturbances in the diet, rather than obesity, underpin the cognitive deficits.

In this study, we found that 25 days of unlimited access to a 10% sucrose solution was enough to induce a prediabetic state in both juvenile and adult animals, as they showed higher blood glucose levels following a glucose overload and hypertriglyceridemia. However, in juvenile animals, the impact of sucrose consumption seems to be higher as they also showed increased fasting glycemia in the short term and maintained the glucose intolerance in the long term. In accordance, it was previously shown that plasma insulin levels remained elevated after access to sucrose ceased in adolescent rats, but not in adult rats [10].

Sucrose-induced alterations in glucose metabolism may contribute to deficits in cognitive performance as fasting blood glucose levels negatively correlated with the percentage of time that rats spent exploring the novel object. Thus, the higher the blood glucose levels, the greater the impairment in cognitive behavior. We even found that blood glucose levels recorded 25 days prior to the NOR test were quite accurate in predicting long-term cognitive performance. Similarly, a previous study shows that consumption of a 32% sucrose solution for 8 weeks impairs NOR and that NOR performance is negatively correlated with BG levels (1 h ITI) [37]. These data may also explain why the sucrose-exposed juvenile group performed poorly in long-term recognition memory, as this group exhibited persistent alterations in glucose metabolism.

Hyperglycemia increases the advanced products of glycation (AGE) and its receptor activation (RAGE) [22,23,24,25]. A recent study shows that NOR deficits induced by high- fructose consumption correlate with increased plasmatic levels of AGEs, oxidative stress, and altered mitochondrial dynamics, especially in the PFC [30].

Here, we found that RAGE levels increased in the mPFC in the sucrose-exposed adult group. However, these animals did not show long-term memory disturbances and RAGE levels did not correlate with NOR performance, implying that the increased RAGE expression is not associated with deficits in cognition at least at this time point. More studies are needed to better characterize the long-term redox status induced by sucrose consumption, such as the measurement of plasma levels of AGEs, TBARS (Thiobarbituric acid reactive substance, a marker of lipid peroxidation), and the expression of antioxidant enzymes (glutathione peroxidase, superoxide dismutase, and catalase) in HIP and mPFC.

BDNF is initially synthetized as a precursor form—the proBDNF that undergoes proteolytic cleavage to become a mature molecule (mBDNF) [67]. A sensitive balance occurs between proBDNF and mBDNF for physiological and pathological conditions [68] and the proBDNF/mBDNF ratio determines the resultant neuronal activity [67]. In the present study, proBDNF levels decreased in animals from the sucrose-exposed juvenile group, whereas it increased in animals from the sucrose-exposed adult group. In addition, HIP proBDNF expression positively correlated with NOR cognitive performance in the hippocampal-dependent trials, T3 and T4, but not in perirhinal dependent T2 trial. We hypothesize that increased proBDNF levels may be related to increased mBDNF in the HIP, which may result in better memory performance, as previously shown in other experimental models [69]. In juvenile and middle-aged rats, proBDNF correlated positively with mBDNF in the HIP, and these rises were accompanied by improved recognition memory [17]. proBDNF itself also improved spatial learning in the HIP [70] and spatial learning itself increased the expression of proBDNF with a corresponding increase in mBDNF [70,71]. Note that in our study, HIP proBDNF levels negatively correlated with the fasting glucose levels. Taken together, these data suggest that sucrose-induced disturbances in glucose metabolism may contribute to decreased proBDNF expression in the HIP, which in turn, likely decreases the recognition memory. This is one possible mechanism by which sucrose would be affecting cognitive performance.

In line with our results, reduced mBDNF expression has also been associated with deficits in cognitive behavior in animals subjected to hypercaloric diets [12,30,72]. Even more, a high-fat, high-sugar diet resulted in impaired performance on the Morris water maze, which correlates with reductions of hippocampal BDNF levels [72]. Further studies are needed to elucidate whether mBDNF is also altered in the NOR impairments in the sucrose-exposed juvenile group.

In summary, the results of this study show that unlimited access to SSBs during youth impacts the developing brain in a different way and with long-lasting consequences compared with the mature adult brain. Specifically, here we found that sucrose exposure during youth worsens the parameters of glucose metabolism in the short and long term (BG and AUC) and decreases HIP proBDNF in the long term, in opposition to sucrose exposure during adulthood. Other studies have also shown that SSBs consumption during youth increases fasting plasma insulin levels and the protein expression of the pro-inflammatory cytokine IL-1b in the liver, whereas the consumption of SSBs during adulthood does not produce such effects [10,39].

Furthermore, here we found that the increased glycemia correlated with a decreased proBDNF expression and poor cognitive performance, implying that the glucose metabolic disturbances are associated with the memory alterations observed in adult animals exposed to sucrose during youth.

In conclusion, the physiological consequences of sucrose overconsumption are not only increasing the prevalence of obesity but also the development of cognitive dysfunction and memory deficits. Our data, along with the growing emergence of other studies, provide evidence that chronic consumption of SSBs in children and adolescents is detrimental to normal mental development and could contribute to the appearance of future cognitive disorders.

## 5. Conclusions

Free access to sugary beverages during youth is detrimental to mental health and can produce long-lasting cognitive deficits in adulthood. In this study, we found that juvenile rats exposed to 10% sucrose show impaired short- and long-term recognition memory (T2 and T3), as well as consolidated memory (T4) in adulthood. Animals that consume sucrose develop hypertriglyceridemia and glucose intolerance. However, sucrose-exposed juvenile animals also show higher fasting glycemia and longer-sustained glucose intolerance. In addition, basal glycemia was negatively correlated with NOR cognitive performance and proBDNF levels. Taken together, these results suggest that altered glucose metabolism and decreased expression of proBDNF in the hippocampus underlie the cognitive deficits observed in adult animals exposed to sucrose during youth.

## Figures and Tables

**Figure 1 biomedicines-10-02723-f001:**
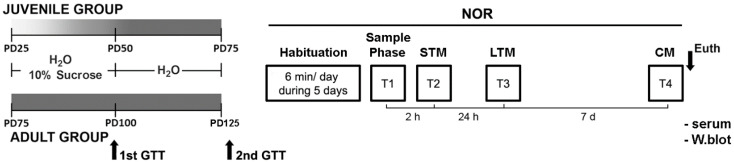
Time line of experimental procedures. The transition zone (soft gray) represents the early youth transition to adulthood (solid color). Over a 25 day period, 25 day old rats (juvenile group) and 75 day old rats (adult group) had unrestricted access to tap water (control group) or the choice to drink a 10% sucrose solution (*w*/*v* made up of tap water, sucrose group) ad libitum. After this period, animals were subjected to GTT and continued to drink only water for other 25 days. Animals were then subjected to NOR and a second GTT. During NOR studies, animals were allowed to explore the objects for 6 min in all the sessions. STM, short term memory (T2); LTM, long term memory (T3); and CM, consolidated memory (T4). The schematic diagram shows the sequential order of the memory task. After NOR, 5 animals were randomly assigned for Western blot (W. blot).

**Figure 2 biomedicines-10-02723-f002:**
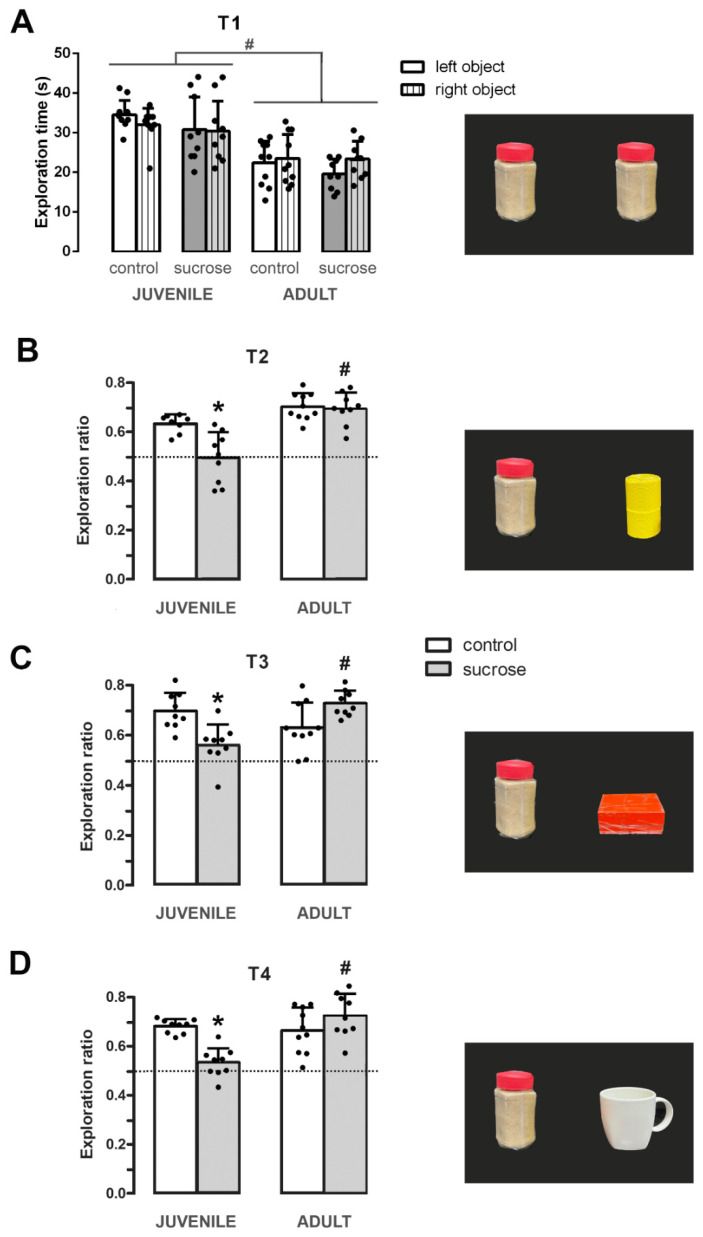
Long-term effect of unlimited sucrose consumption on learning ability. The NOR task was performed 25 days after sucrose treatment both in juvenile and adult animals. (**A**) Time spent exploring the objects during the sample phase (T1). (**B**–**D**) Exploration ratios during the choice phase of the NOR trials (T2–T4). Juvenile or adult rats exposed to only water (control, white bars) or 10% (*w/v*) sucrose (sucrose, gray bars). Note that adult animals from the sucrose-exposed juvenile group show impaired acquisition and consolidation of the object trace. Data are presented as mean ± SD from two independent experiments, n = 9–10. Tukey’s post hoc test showed differences by treatment, * *p* < 0.05 and by age, # *p* < 0.01 juvenile group vs. adult group (T1), and sucrose-exposed juvenile group vs. sucrose-exposed adult group (T2–T4). Dotted lines show the exploration rate at the value 0.5.

**Figure 3 biomedicines-10-02723-f003:**
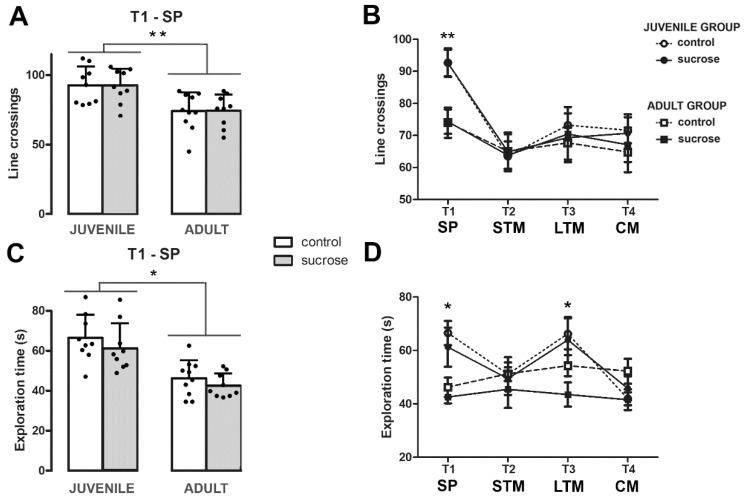
(**A**) Locomotor activity during the familiarization session and (**B**) during T1–T4 NOR trials. (**C**) Total time spent exploring the objects during the familiarization session and (**D**) during the T1–T4 NOR trials. No significant differences in locomotor activity or the total exploration time were found between the control and sucrose groups by two-way ANOVA and two-way RM ANOVA. SP, sample phase (T1); STM, short-term memory (T2); LTM, long-term memory (T3); and CM, consolidated memory (T4). Data are presented as mean ± SD from two independent experiments, n = 9–10. Tukey’s post hoc test showed only differences by age, ** *p* < 0.005, and * *p* < 0.01.

**Figure 4 biomedicines-10-02723-f004:**
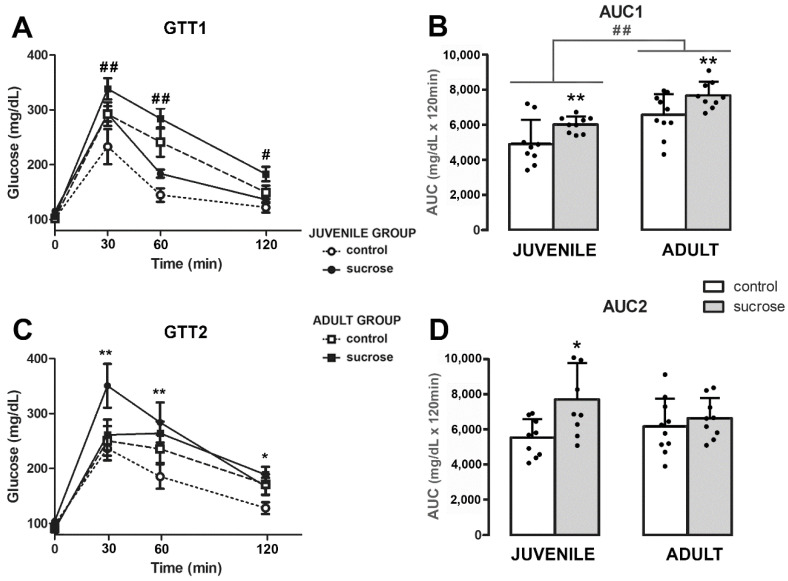
(**A**) Glycemia levels during a GTT performed 25 days after the initiation of sucrose treatment in juvenile (at 50 days-old, circles) and adult (at 100 days old, squares) animals. Controls (open circle/square) and animals exposed to sucrose (filled circle/square). (**B**) Values of the AUC. (**C**) Second GTT performed 25 days after finishing the sucrose treatment (75 days old for the juvenile group and 125 days old for the adult group) and (**D**) their respective AUC values. Data are presented as mean ± SD from two independent experiments, n = 9–10. Tukey’s post hoc test showed differences by treatment, ** *p* < 0.01, * *p* < 0.05, and by age ## *p* < 0.001, # *p* < 0.05.

**Figure 5 biomedicines-10-02723-f005:**
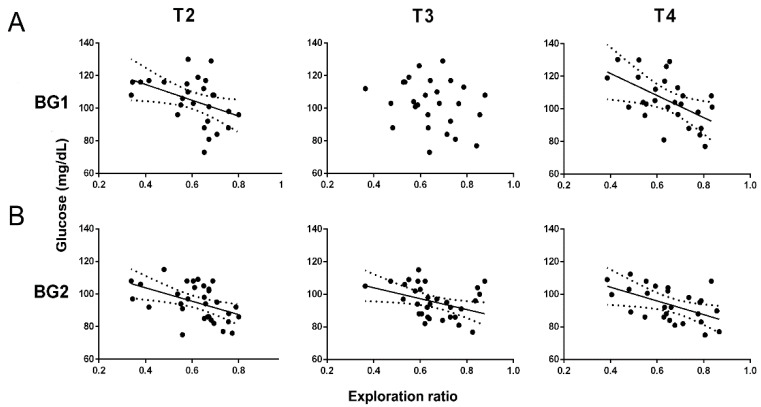
(**A**) Correlations between BG levels 25 days after initiation of sucrose treatment (BG1) and the NOR exploration ratios T2, T3, and T4. Correlation equations of T2, y = 134.1 − 48.54x and T4, y = 148.8 − 67.59x. (**B**) Correlations between the BG levels 25 days after finishing the sucrose treatment (BG2) and the NOR exploration ratios T2, T3, and T4. Correlation equations of T2, y = 120.3 + 41.10x; T3, y = 117.4 + 33.47x and T4, y = 121.2 + 42.08x. Each point represents a value corresponding to an individual animal in the control and sucrose groups of both ages (n = 8). Dotted curves represent the 95% confidence intervals.

**Figure 6 biomedicines-10-02723-f006:**
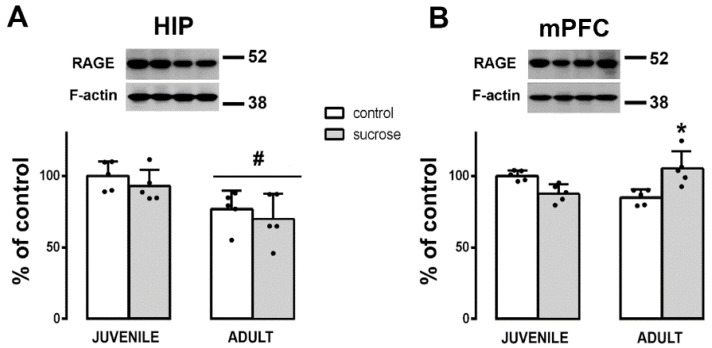
(**A**) Western blot of RAGE in the (**A**) HIP and the (**B**) mPFC of control (white bars) and sucrose-exposed (gray bars) animals from juvenile or adult groups. Data were quantified by densitometric analysis and corrected with reference to the F-actin loading control. Representative pictures of RAGE expression and the F-actin loading control are shown in the upper panel. Data are presented as mean ± SD from two independent experiments, n = 5. Tukey’s post hoc test showed differences by treatment, * *p* < 0.05 and by age # *p* < 0.005.

**Figure 7 biomedicines-10-02723-f007:**
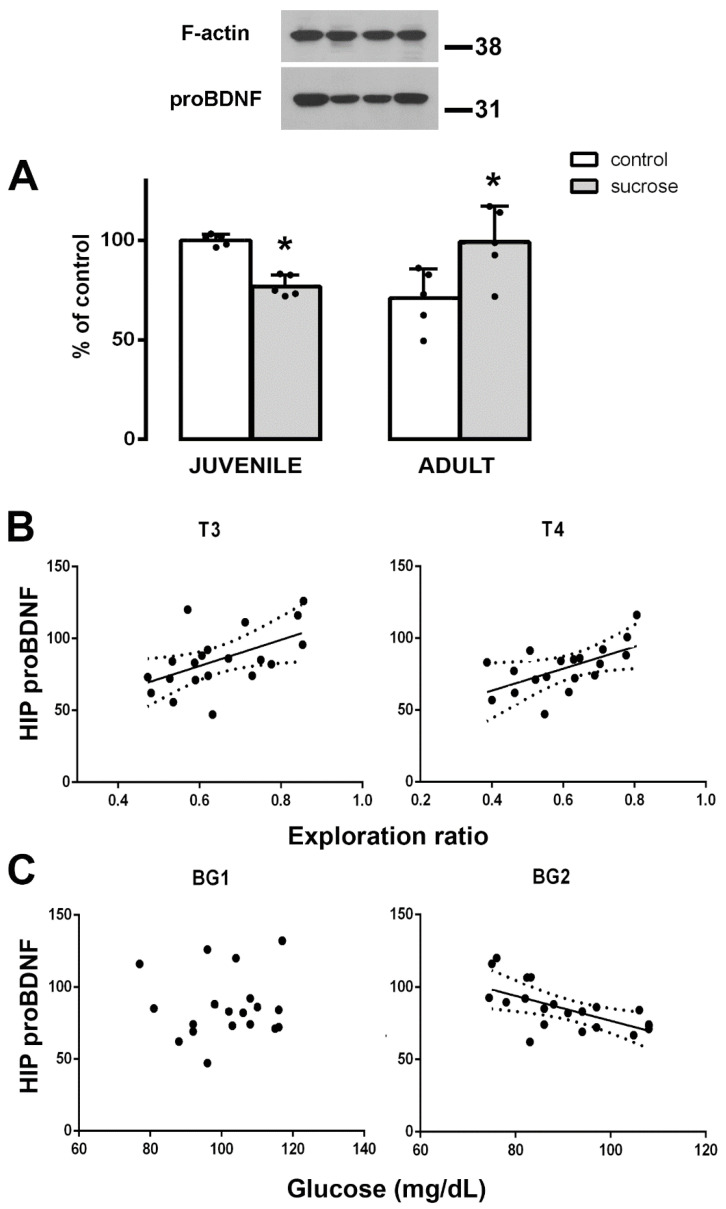
(**A**) Western blot of proBDNF in the HIP of control (white bars) and sucrose-exposed (gray bars) animals from juvenile or adult groups. Data were quantified by densitometric analysis and corrected with reference to the F-actin loading control. Representative pictures of proBDNF expression and the F-actin loading control are shown in the upper panel. Data are presented as mean ± SD from two independent experiments, n = 5. Tukey’s post hoc test, * *p* < 0.05. (**B**) Correlations between the hippocampal proBDNF levels and the NOR exploration ratios T3 and T4. Correlation equations of T3, y = 26.33 + 90.88x and T4, y = 32.88 + 76.35x. (**C**) Correlations between the hippocampal proBDNF levels and BG1/BG2. Correlation equation of BG2, y = 163.113 − 0.864x. Each point represents a value corresponding to an individual animal in the control and sucrose groups of both ages. Dotted curves represent the 95% confidence intervals.

**Table 1 biomedicines-10-02723-t001:** Plasmatic levels of BG, TG, and corticosterone (COR) at the end of the 25 days of sucrose treatment and 25 days after finishing the sucrose treatment. Data are presented as mean ± SD from two independent experiments, n = 8–10. Tukey’s post hoc test showed differences by sucrose treatment, ** *p* < 0.005.

	Juvenile Group				Adult Group			
	PD50		PD75		PD100		PD125	
	Control	Sucrose	Control	Sucrose	Control	Sucrose	Control	Sucrose
BG (mg/dL)	101 ± 12	115 ± 7 **	98 ± 12	104 ± 8	104 ± 19	105 ± 24	92 ± 9	89 ± 4
TG (mg/L)	136 ± 40	303 ± 103 **	117 ± 28	116 ± 44	127 ± 39	259 ± 79 **	124 ± 31	129 ± 45
COR (ng/mL)			15 ± 8	15± 5			14 ± 7	23 ± 9

## Data Availability

The datasets used and analyzed during the course of this study are available from the corresponding author upon reasonable request.

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
