# Peer review of "Long-Term Memory Function Impairments following Sucrose Exposure in Juvenile versus Adult Rats"

_biomedicines, 2022, doi:10.3390/biomedicines10112723_

Round 1

Reviewer 1 Report

1.      In the abstract, when the authors mention their NOR results, they said "However, when the four groups were exposed 2, 24 h or 7 days later to a copy of the objects presented in T1 and a novel object, control animals or animals that drank sucrose during the adulthood, but not during youth, exhibited a clear preference to explore the novel object" (lines 17-19). However, the interesting data here is that animals that drank sucrose during the youth develop memory deficits, and it is that which should be highlighted in the abstract. 

2.      In the graphical abstract legend, the authors should explain what the “GTT curve” (line 32) is.

3.      In the introduction, when authors said: “showed by an increased microglia” (line 49) they should explain more detailly which microglia parameters are increased (e.g. proliferation, migration, activation, particular markers, etc.).

4.      In line 53: “and its precursor, proBDNF, are key modulators” it should be said “and its precursor, proBDNF, is a key modulator

5.      In line 55: “Proliferation of neuronal progenitor cells occurs in the dentate gyrus (DG) of the HIP” is not precise information. Authors are speaking about adult neurogenesis, and they should specify it here and in the rest of the paragraph and in every place in the article when neurogenesis is regarded. Moreover, DG is not the only brain area where there is adult neurogenesis, so authors should at least mention other(s).

6.      To said what authors say in lines 61-63 “Altogether these studies suggest that high-sucrose diets mechanisms to impair hippocampal neuroplasticity may include increases of pro-inflammatory cytokines, as well as decreases of neurotrophins, which may ultimately contribute to diet-induced cognitive deficits” they should cite some references in which deleterious effects of increased pro-inflammatory cytokines and decreased BDNF on hippocampal plasticity and NOR memory have been demonstrated (as they have been).

7.      In Material and Methods 2.1. Experimental animals section, it should be justified why authors chose 10% sucrose and not higher or lower concentrations, which is of special interest because they stablished the importance of the study because of human SSB consumption and these drinks have far more sugar amounts than 10%.  

8.      In lines 174-175 authors explain that they measured exploration ratio in NOR as the time spent exploring the novel object divided by the time spent exploring both objects [tN/ (tN + tF)]. However, classically he discrimination ratio in this kind of paradigms is calculated as the difference in time spent by each animal exploring the novel compared with the familiar object divided by the total time spent exploring both objects: (tN -tF) / (tN + tF). Authors should consider this discrepancy and explain why they calculate this ratio and the implications it could have in comparing with previous studies.

9.      In line 183 authos say “A total of 2 0mg of protein was separated by 10% SDS–PAGE” and I assume it should be 20μg instead, since 20mg is a disproportionately huge amount of protein

10. The authors do not explain why the study protein expression in the mPFC. Hippocampus and perirhinal cortex are said in the article to be related with NOR performance but not mPFC. If so, it should be justified more clearly.

11.  In the statistics section of Methods authors should justify why they choose two-way ANOVA, three-way ANOVA or two-way repeated measures ANOVA explaining the factors implicated in each case.

12. Regarding the NOR results (figure 2), sucrose increases exploration ratio in animals exposed to sucrose during the adulthood and T3 and T4 and these results are not properly described.

13.  Probably the most improvable aspect of the article regards to the neuroinflammatory aspects. RAGE is a very vague and unspecific marker so its relationship with neuroinflammation is not as solid as the authors claim. To strength the history told here, neuroinflammation should be measure by more solid markers such as IL-1β or TNF-α levels or microglia activation/proliferation assays. 

Author Response

We have carefully reviewed the manuscript based on your comments and we think that the changes made have improved the quality of the manuscript.

Reviewer 2 Report

In this manuscript, authors investigated the short term and long-term effects of sucrose consumption on the memory function in male rats at different ages (young and adult). They found differential changes between two ages. There are several issues in the manuscript which makes it difficult to understand the aim and the results.

1.     In title, authors mentioned the name of the behavioral test for testing memory function, however it is better just to mention memory function.

2.     In the abstract there is no information about the significance of study scientifically which means that it is not clear what is the scientific question.

3.     Result section in the abstract is very confusing, there is no information about memory function in the young rats,

4.     It is important to use the same definition for the whole manuscript, juvenile or youth.

5.     Why authors included only male rats?

6.     Authors had 10 animals per group (control and sucrose) and then they were divided into two groups ( 4-5 animals per group), this number is low to get reliable results , how authors calculated the power of the study design? It is important to add the information about power analysis.

7.     For the NOB test, it would better to add the pic of used objects.

8.     What was the criteria to exclude the animals in the T1 phase?

9.     It is important to have illustration of NOB for 4 phases which is confusing in the text. It is not clear if authors changed one object for each phase from T2 to T4?

10. What is the meaning of counting cells in lower blades of the granular cell layer (GCL) of the DG?

11. Which objective lens was used to count the cells?

12.For 70-um thickness, how authors made sure about penetration of Ab?

13.At which thickness counting of cells was performed?

14. In the presented formula: how authors calculated cell density (N), if V is the volume why they added thickness in the formula separately? How many sections were used to measure the volume of area? Was there any significant difference in the volume of region which can affect the number of cells.

15.What was the criteria for counting the cells?

16. In the statistical section, did authors test normal distribution of data? And also, homogeneity of data?

17. Bar chart is not a good option for showing the differences between the groups, it is needed to be changed to dot plot to see the distribution of data. For example, in Fig 3.A in adult group authors added ** while the value in two groups are the same.

18.Fig 8 which is related to counting the number of cells included in examples of images for counting and especially for DCX , the staining and magnification are not clear for counting the cells which makes it difficult to interpret the results.

Author Response

(The authors gave the same response as above.)

Reviewer 3 Report

In the paper of Coirini et al., the authors showed that prolonged consumption of sucrose in young rats leads to a delayed deterioration in the long-term object recognition memory. This effect was correlated with a decrease in neuronal proliferation in the hippocampus and the level of precursor for brain-derived neurotrophic growth factor (pro-BDNF). The experimental design is well thought out and the results obtained are very interesting and shed light on the long-term harms of excess sugar consumption at an early age. I do not have any fundamental remarks, there are minor issues for discussion:

1) If I understood correctly, then the presented objects during NOR task, in addition to having different textures and shapes, were also of different colors. "Accordingly, the rats were re-exposed to two objects for 6 min: a copy of the familiar (F) object and the novel (N) object (i.e. one glass jar was exchanged for a rough yellow metal cylinder 9.5 cm high and 5.5 cm in diameter, an orange plastic box 10.5 cm long x 9 cm wide x 4 cm high and a conical cup of white ceramic 8.5 cm high and 8.5 cm in diameter at T2, T3 and T4, respectively)", 'Materials and methods' section. It is known that rats have only two types of photoreceptor cones, which limit the perception of color vision. Could this somehow affect the results of object recognition testing? Please discuss it.

2) Theoretically, the time an animal spends near a recognizable object may depend on its visual acuity. Since animals of different ages were used, it would be a good idea to discuss whether the vision in the older age group (adult) is worse than in young rats (juvenile).

3) According to the Fig. 3A and 3C, the locomotor activities of juvenile and adult rats were differed. With what it can be connected?

4) Figure 7 contains data on the precursor for brain-derived neurotrophic factor (pro-BDNF). These data may not precisely reflect the level of the final product (BDNF). Because the precursor protein can be eliminated, for example, proteasome degradation without BDNF translation. Thus, an increase in the expression of pro-BDNF in adult rats does not yet indicate that the level of translation of BDNF itself increases. Please discuss it.

5) The format of references needs to be adjusted according to the requirements of the journal.

6) The manuscript needs some correction of the lexicon. For example, instead of 'sections' [of hippocampus], it is better to write 'slices' [of hippocampus].

Author Response

(The authors gave the same response as above.)

Round 2

Reviewer 1 Report

All the changes made have contribute to improve the quality of the masnucript and the presentation of the results. However, I have still concerns regarding neuroinflammation. I am aware that RAGE results do not correlate with other results here but if the authors mention in the introduction the contribution of pro-inflammatory cytokines, the logical poing to me is to measure them. Otherwise, I would not mention them at all. Therefore, from my point of view there are two options: 1) to carry out some easy and fast measurement of pro-inflammatory cytokines (e.g. ELISA), which is what I strongly recommend, or 2) to change the neuroinflammation alusions in the article to speak only about RAGE and not parameters with have not been measured

Author Response

We have accepted de second option and removed the neuroinflammation allusions in the manuscript.

Reviewer 2 Report

Authors responded to the comments in a satisfactory manner. The only thing is that as authors agreed, it would be great if they remove the entire section 3.5 of the manuscript, since the method of counting and the images do not show reliable data collection. 

Author Response

We agree to remove the section 3.5 as you suggested.

Round 3

Reviewer 1 Report

No comments since you have accepted de second option and removed the neuroinflammation allusions in the manuscript.